# Microenvironmental pH Modification in Buccal/Sublingual Dosage Forms for Systemic Drug Delivery

**DOI:** 10.3390/pharmaceutics15020637

**Published:** 2023-02-14

**Authors:** Shaolong He, Huiling Mu

**Affiliations:** 1Qilu Pharmaceutical (Hainan) Co., Ltd., Haikou 570311, China; 2Department of Pharmacy, Faculty of Health and Medical Sciences, University of Copenhagen, Universitetsparken 2, DK-2100 Copenhagen, Denmark

**Keywords:** absorption enhancement, microenvironmental pH modification, buccal/sublingual dosage form, solubility

## Abstract

Many drug candidates are poorly water-soluble. Microenvironmental pH (pH_M_) modification in buccal/sublingual dosage forms has attracted increasing interest as a promising pharmaceutical strategy to enhance the oral mucosal absorption of drugs with pH-dependent solubility. Optimizing drug absorption at the oral mucosa using pH_M_ modification is considered to be a compromise between drug solubility and drug lipophilicity (Log D)/permeation. To create a desired pH_M_ around formulations during the dissolution process, a suitable amount of pH modifiers should be added in the formulations, and the appropriate methods of pH_M_ measurement are required. Despite pH_M_ modification having been demonstrated to be effective in enhancing the oral mucosal absorption of drugs, some potential risks, such as oral mucosal irritation and teeth erosion caused by the pH modifiers, should not been neglected during the formulation design process. This review aims to provide a short introduction to the pH_M_ modification concept in buccal/sublingual dosage forms, the properties of saliva related to pH_M_ modification, as well as suitable drug candidates and pH modifiers for pH_M_ modifying buccal/sublingual formulations. Additionally, the methods of pH_M_ measurement, pH_M_ modification methods and the corresponding challenges are summarized in the present review.

## 1. Introduction

Buccal/Sublingual administration is an attractive route to achieve systemic drug delivery. It offers advantages such as circumventing the hepatic first-pass metabolism and chemical/biological drug degradation associated with oral administration, the quick onset of drug action, ease of administration and relatively high level of patient compliance [1,2,3,4]. In general, formulations such as mucoadhesive films, patches, tablets, gels, etc., have been employed for buccal/sublingual drug delivery, which could increase the contact time between drugs and the oral mucosa. Among the various physicochemical properties of the drug candidates, aqueous solubility and lipophilicity (Log D)/permeation are two crucial factors affecting drug absorption at the oral mucosa. A good drug candidate should be soluble in human saliva, and possess enough lipophilicity to permeate the epithelium at the oral mucosa [3]. However, it is estimated that over 40% of drugs on the market are poorly water-soluble [5], which could lead to a slow drug release. Generally, poorly water-soluble drugs are weakly ionizable drugs, which might have pH-dependent solubility and/or pH-dependent lipophilicity (Log D)/permeation. Therefore, pH plays a crucial role in the absorption of those drugs in the oral cavity.

Microenvironmental pH (pH_M_) modification is widely used in oral solid dosage forms to increase the dissolution of poorly water-soluble drugs with pH-dependent solubility in the gastrointestinal (GI) tract, and this approach creates a microenvironment with an ideal pH level in the vicinity and inside the solid dosage forms by adding pH modifiers to the formulations [6,7,8,9]. It has been demonstrated that pH_M_ modification is an effective way to increase drug dissolution and enhance drug absorption in the GI tract [7,10,11,12]. Typically, the pH rapidly changes from highly acidic in the stomach to neutral in the small intestine. Under fasted-state conditions, the gastric pH range is between 1.7 and 4.7, and the pH in the small intestine slightly increases from 5.9 in the proximal parts to 7.8 in distal parts for the human subjects [13]. The physiological environment for drug dissolution in the oral cavity is different compared to that in the GI tract. The pH range for unstimulated human saliva is 6.2 to 7.6 and the average salivary pH is 6.8 [14,15]. The neutral pH environment, without dramatic shifts, is beneficial to create a desired pH_M_. Additionally, the limited volume (average 0.8 to 1.1 mL) and the low secretion rate of human saliva (0.35 to 2.00 mL/min) [16,17] could lead to a slow release of the pH modifiers from the formulations and, hence, to maintaining the ideal pH_M_. Therefore, the oral cavity provides a suitable physiological environment for pH_M_ modifications. pH_M_ might not only affect drug release from formulations, but also drug permeation across the oral mucosa. Generally, pH modifiers could modulate pH_M_ both in and in the vicinity of the buccal/sublingual formulations and affect the drug ionization, and thereby influence the drug permeation across the oral mucosa. Thus, pH_M_ modification in buccal/sublingual dosage forms might be an effective strategy to enhance the absorption of drugs with pH-dependent solubility or/and permeation.

The aim of this review paper is to provide an overview of pH_M_ modification in buccal/sublingual dosage forms for systemic drug delivery. In this article, the concept of pH_M_ modification in the oral cavity is discussed. The drug candidates, pH modifiers and pH_M_ measurement methods are summarized. Most importantly, different types of pH_M_ modification and the corresponding cases, as well as their challenges, are highlighted. We hope that this review will be useful for the future design and development of pH_M_ modifying buccal/sublingual formulations.

## 2. Concept of Microenvironmental pH (pH_M_) Modification in the Buccal/Sublingual Dosage Forms

pH_M_ modification is a pharmaceutical strategy to create a microenvironment with a targeted pH in or/and around formulations during dissolution by adding pH modifiers to formulations. In this review, we will present and discuss the theories associated with pH-dependent drug dissolution/release and pH-dependent drug permeation, as well as pH_M_ modification in buccal/sublingual dosage forms.

### 2.1. Theory: pH-Dependent Dissolution and Permeation

Drug dissolution/release from buccal/sublingual formulations is one of the crucial factors affecting drug absorption at the oral mucosa. The relationship between the pH, drug solubility and dissolution rate has been elucidated using the Nernst-Noyes-Whitney equation [18] (Equation (1)) and the “solubility-pH” equations (take monoacidic drugs and monobasic drugs as examples) [19,20] (Equations (2) and (3)), as described below:(1)$$\frac{dC}{dt}=\frac{DS}{Vh}\;(Cs_{\;}-C_{b})$$
where $\frac{dC}{dt}$ is the dissolution rate, *D* is the diffusion coefficient, *S* is the surface area of solid exposed, *V* is the volume of dissolution media, *h* is the thickness of the diffusion layer, *C_s_* is the concentration (saturated) of drug at the solid surface and *C_b_* is the concentration of drug in the bulk medium.
(2)$$C_{s}=C_{S0}\;[1+10{\;}^{\mathrm{pH}\;-pKa}]\;(\mathrm{for}\;\mathrm{monoacidic}\;\mathrm{drugs})$$
(3)$$C_{s}=C_{S0}\;[1+10^{\;pKa\;-\mathrm{pH}}]\;(\mathrm{for}\;\mathrm{monobasic}\;\mathrm{drugs})$$
where *C*_s_ is the drug solubility at a given pH and *C*_S0_ is the intrinsic solubility of the drug.

According to the “Solubility-pH” equations, a slight shift in the pH might lead to a significant change in the drug solubility. Theoretically, decreasing the pH could improve the solubility of a weakly basic drug by increasing the concentration of ionized drug in the solution. When most of the dissolved drug substance remains in its ionized form, a further decrease in the pH has little effect on its solubility, and the drug solubility approaches a plateau level in the pH-solubility profile. A suitable pH level at the surface of a solid formulation exposed to dissolution media could increase the local drug concentration (*C*_s_) and, consequently, enhance the drug dissolution and release ($\frac{dC}{dt}$) from the solid formulation.

The mechanism of drug transport across the oral epithelium is similar to that across the other epithelia in the human body. Generally, both the transcellular and paracellular pathways are involved in this process [3,21,22,23,24]. For the drugs transported mainly via the transcellular route, drug permeation across the oral mucosa might be affected by the pH at the oral mucosa. According to the pH-partition theory, the neutral forms of drugs are more permeable (lipophilic) than the ionized species; therefore, a pH shift not only affects the dissociation of weakly ionizable drugs, but also the drug permeation across biological membranes (Figure 1) [25,26].

### 2.2. pH_max_ Concept

A pH_max_ concept defined as the pH value at which a given drug has a maximal aqueous solubility and the sum of its ionized species and unionized species in solution is at a maximum [27]. Modulating the saliva pH at the sublingual mucosa to the pH_max_ by adding buffering agents in sublingual formulations was expected to lead to a maximal potential absorption. The pH_max_ concept was proven to be valuable in the case of a propranolol sublingual tablet with buffering agents, which achieved a higher absorption in human subjects than the conventional non-buffered tablet. However, previous studies regarding the metroprolol buccal tablet and gel did not support the pH_max_ concept. A specific pH level (rather than pH_max_) also led to the highest buccal absorption of metroprolol [28,29].

### 2.3. Microenvironmental pH Modification in Buccal/Sublingual Dosage Forms

The aim of pH_M_ modification is to enhance the absorption of a given drug by influencing its solubility and permeability. Upon pH_M_ modification in buccal/sublingual dosage forms, a small space exists between the formulation and the mucous membrane when the formulation is attached the oral mucosa. Typically, a drug must have a sufficient aqueous solubility to be released from the formulation and dissolved in the space, before permeating through the membrane of the oral mucosa. The pH of the space and the pH inside the formulation could be modified by adding pH modifiers into the formulation, which might affect the drug release (caused by the changes in drug solubility), drug solubility in the space and drug permeation across the mucosa by influencing the drug dissociation (Figure 2). In general, the drug, pH modifier and mucoadhesive polymer are the main components of the pH_M_ modifying buccal/sublingual formulations.

## 3. Properties of Saliva Associated with pH Modification

The main functions of saliva are to maintain oral health and help to build and maintain the health of hard and soft tissues. Approximately 99% of saliva is water, and the other 1% consists of a variety of electrolytes and proteins [30,31]. Regarding buccal/sublingual drug delivery, saliva provides a water-rich environment that facilitates in the drug dissolution and release from buccal/sublingual formulations before the drugs permeate through the membrane of oral mucosa [32]. To achieve a successful pH_M_ modification in buccal/sublingual formulations, some properties of saliva should be taken into consideration during the formulation design.

### 3.1. pH and Buffer Capacity of Saliva

Human saliva has been reported to have a pH range of 6.2–7.6 [14]. The composition of saliva secreted from different regions differs, leading to different saliva pH. The pH in the palate, the floor of the mouth, the buccal mucosa and the tongue, in humans, have been reported to be 7.3, 6.5, 6.3 and 6.8, respectively [33]. The flow of saliva with a buffer capacity resisting pH shift could remove acidic and basic foods on the oral mucosa, maintaining the pH in the oral cavity near neutrality in a long term. The bicarbonate, the phosphate and the protein buffer systems in the whole saliva are the major systems contributing to the buffer capacity, and their concentration and buffer capacity are dependent on the secretion rate of saliva [34,35,36]. The bicarbonate system is considered to be the principal buffer of saliva and its dynamics system is complicated. The buffer capacity ranges of unstimulated human saliva and stimulated human saliva not exposed to the atmosphere (in the pH range of 4.25 to 6.75) have been reported to be 1.9–7.7 mmol H^+^/(L saliva × pH unit) and 2.4–9.3 mmol H^+^/(L saliva × pH unit), respectively. A high secretion rate of saliva implies a high concentration of bicarbonate in the saliva, which might lead to a high buffer capacity [36].

### 3.2. Secretion Rate of Saliva and Thickness of Salivary Film

Saliva is a complex mixture secreted by salivary glands. There are three pairs of major glands: the parotid, submandibular and sublingual glands, and numerous minor salivary glands [37]. Salivary secretion continues throughout the day, with an average total volume of 500–600 mL. Previous studies have reported that the mean flow rate of unstimulated human saliva and stimulated human saliva is 0.35 mL/min and 2 mL/min, respectively [16,38]. The function of the salivary glands is under the influence of various stimulations. For instance, tasting, smelling and chewing food can affect the properties, composition, volume and flow rate of saliva [16,37]. Additionally, some physiological factors, such as gender and age, have been reported to affect the salivary flow rate [17,39]. The volume of human saliva before swallowing is around 1.1 mL. Following a swallow, around 0.8 mL saliva stays in the mouth, and much of the saliva is present as a film on the mucosa and the surface of hard tissues in the oral cavity [17]. The estimated thickness of the human salivary films is 70–100 µm (calculated by dividing the volume of saliva by the surface area of oral mucosa) [40,41]. However, the thickness of the salivary film varies at different regions. The thicknesses of the human salivary film on the anterior tongue, buccal surface and anterior hard palate have been estimated to be 50–70 µm, 40–50 µm and 10 µm, respectively, by measuring the wetness of filter paper strips applied to different regions [16,42].

## 4. Drug Candidate and pH Modifier for Buccal/Sublingual Dosage Forms

### 4.1. Drug Candidate

The low drug loading capacity of buccal/sublingual formulations and the limited absorption area in the oral cavity are two main limitations for buccal/sublingual drug delivery. Thus, drug candidates should be high potency to achieve successful therapeutic efficacy. In addition, suitable drug candidates must not cause local irritation and toxicity at oral mucosa. Regarding physicochemical properties, high lipophilicity (log P (octanol/water) > 2), fairly good water-solubility and small molecular size (less than 800 Da) are typically considered as ideal parameters for drug candidates, as described previously [3]. The extent of different drug transport pathways across the epithelium depends on the drug physicochemical properties [43,44]. Typically, drug candidates with high lipophilicity can move across the lipid-rich epithelial cell membrane with relative ease. Fairly good water solubility allows for the fast drug release of buccal/sublingual formulations and drug diffusion across the hydrophilic cytoplasm of cells and paracellular passage. Macromolecules can be delivered via the oral mucosa, e.g., buccal insulin spray (Generex Oral-lyn^®^) was approved by Food and Drug Administration (FDA) for the treatment of patients under the Investigational New Drug (IND) program [45,46,47]. However, the number of marketed buccal/sublingual macromolecules is very small. Most of the marketed buccal/sublingual delivered drugs are small molecules. The selected drug products recorded in the FDA-Approved Drugs database [48] are summarized in Table 1 (only new drug applications are listed in this table and some drug products with the same active ingredients have different strengths or dosage forms).

However, over 40% of marketed drugs and approximately 90% of drug candidates are reported to be poorly water-soluble [5], and most of them are weakly ionizable drugs, indicating that their solubility and/or permeability across the lipid-rich epithelium are pH-dependent [49,50,51,52]. Typically, the ionic form of a drug is more water soluble than its non-ionic form. A change in the pH might influence the ratio of the ionized form of the dissolved drug, according to the Henderson-Hasselbach equation (Equation (4)) [53]. When the difference in the water solubility (and/or lipophilicity) between the two forms is big enough, a slight pH change might have a significant effect on the drug solubility. Therefore, drug candidates suitable for pH_M_ modification should have pH-dependent solubility and/or pH-dependent lipophilicity and be poorly soluble at physiological pH in the oral cavity. The physicochemical properties of some transmucosal delivered drugs (from DrugBank database) and the relevant literature on the buccal/sublingual delivery of the drugs are summarized in Table 2.
(4)$$\mathrm{pH}=pK_{a}+\mathrm{Log}\;\frac{\left[A-\right]}{\left[HA\right]}\;(\mathrm{for}\;\mathrm{monoacidic}\;\mathrm{drugs})$$
where p*K*_a_ is the negative log of the drug dissociation constant; [A−] is the concentration of the base form of the drug; [HA] is the concentration of the acidic form of the drug.

### 4.2. pH Modifier

There are a few concerns about the excipients used in pharmaceutical formulations. A pH modifier can only be considered as a pharmaceutical excipient if it has been demonstrated to be safe for human beings. So far, various pH modifiers have been applied in the food and pharmaceutical industries. The Generally Recognized as Safe (GRAS) list of the FDA lists some safe pH modifiers that have been added to food. In addition, various pH modifiers recommended for oral liquids have been collected in the United States Pharmacopeia (USP). However, the specific pH modifiers for buccal/sublingual formulations were not referenced. The pH modifiers collected in the USP [67] and their maximum potency per unit dose used in solid oral and buccal/sublingual formulations in the database of Inactive Ingredient Search for Approved Drug Products Search, provided by the FDA [68], are summarized in Table 3. The pH modifiers can be divided into three categories: acidifying agents, alkalizing agents and buffering agents. Currently, only a few pH modifiers, as shown in Table 3, were applied in the commercial buccal/sublingual formulations approved by the FDA. pH modifiers demonstrated without local irritation and toxicity to oral mucosa could also be potential choices for the buccal/sublingual dosage forms.

## 5. Methods for Microenvironmental pH Measurement

Typically, it is easy to determine the pH of pharmaceutical solutions potentiometrically. Analyzing the pH_M_ in the vicinity of pharmaceutical solids or in the matrix of formulations during the drug dissolution and release process is much more challenging. Several techniques have been applied to gain information on the pH_M_, and to investigate the relationship between the pH_M_ and drug dissolution behavior. pH-indicating dyes have been used to determine the pH_M_ within and around the tablets during hydration [7,69,70]. This method only roughly estimates the pH_M_ according to the relationship between the pH level and the color of dyes. In addition, the pH electrode has been employed to precisely investigate the pH_M_ in the concentrated suspension of pharmaceutical solids and on the surface of hydrated tablets [71,72]. To provide the data of a quantitative nature and detailed insights into pH effects during drug dissolution and release process, fluorescence imaging [73,74], the UV/Vis imaging method [75,76] and electron paramagnetic resonance (EPR) imaging [62] have been applied to non-invasive image pH sensitive fluorescent agents, pH-indicating dyes and pH-sensitive paramagnetic compounds, respectively, giving spatial resolutions of pH_M_. Generally, in buccal/sublingual formulations, the surface pH (pH_M_) in the vicinity of the formulations during the swelling and dissolving process have been measured, and several methods are presented in the following section.

### 5.1. pH Electrode Approach

The most common method to determine the pH_M_ is the pH electrode approach. As the previous studies described [57,77,78,79,80,81,82], formulations (e.g., tablets, film and patch) were allowed to swell in a limited volume of buffer solution (at neutral pH) at room temperature for a certain period. Subsequently, the pH on the surface of the formulations was determined using a pH electrode. Mucoadhesive buccal films containing ornidazole were allowed to swell in 4 mL of phosphate buffer (pH 6.8 ± 0.1) at room temperature for 120 min and the surface pH was measured using an electrode pH meter [77]. A relatively short time and low volume of medium was needed for the swelling of the rapidly dissolving films of naftopidil before the pH measurement [57]. For the polymer-based buccal tablets with a long retention time, the time period for swelling was around two hours [79,83]. Although this method is easy to operate, it is challenging to investigate the changes in the pH_M_ adjacent to the surface of the formulations during the initial swelling process.

### 5.2. Computer-Enhanced Color Images Method

To gain more information on the pH_M_ change during the dissolving process of the fentanyl tablet, a computer-enhanced color images (of pH paper) method was used to record the pH_M_ as it varied over the surface of the swelling tablet [54]. The schematic view of the setup and the computer-enhanced color images of pH paper are shown in Figure 3. A piece of pH paper was placed over a tablet. The tablet with the pH paper was held between two microscope slides, and a small volume of deionized water was applied to the pH paper. The tablet was rapidly wetted by the water that permeated the pH paper. As the tablet swelled, the pH paper was digitally photographed at different time intervals. The pH over the distinct regions of the tablet surface were then determined from the digital images and in comparison to the reference pH standards. The pH_M_ decreased from 7.0 to 5.0, and then gradually increased to around 6.0 during the first 5 min of the dissolving process [54].

### 5.3. UV/Vis Imaging Method

In one of our previous works, an UV/Vis imaging method with an agarose hydrogel mimicking the fluid on the surface of the buccal mucosa was constructed. The effect of the malic acid dose on the pH_M_ during the initial dissolution of the buccal films, and the information related to film the swelling and possible drug precipitation in the films were monitored using this method [58]. The schematic view of the UV/Vis imaging setup is shown in Figure 4. The agarose hydrogel contained agarose (0.5% *w*/*v*), bromothymol blue (pH indicator, 6.29 × 10^−5^ M) and a buffer solution, simulating the human saliva pH and buffer capacity. A buccal film was attached on the agarose hydrogel, and the absorbance change of the pH indicator in the hydrogel at a wavelength of 610 nm was monitored during the swelling of the film. To relate the absorbance of bromothymol blue to the pH in the hydrogel, an absorbance-pH profile was constructed as a calibration curve. Based on the calibration curve, the pH during the swelling of the buccal film could be measured. The pH_M_ in the vicinity of the buccal films at different time points were determined and selected images are shown in Figure 5. The addition of malic acid in the films led to an obvious decrease in the pH_M_ at 5 min, whereas the pH_M_ was increased due to the release of saquinavir from the films at 8 min (Figure 5B,C).

## 6. Microenvironmental pH (pH_M_) Modification Methods

According to the pH modifier classification described in Section 4.2, formulating acidifying/alkalizing agents and buffer agents are two common methods for pH modification. In addition, the application of effervescence in formulations constitutes another strategy for changing the pH_M_, which facilitates drug release from the formulations [54,84]. These pH_M_ modification methods have been employed in buccal/sublingual dosage forms.

### 6.1. Microenvironmental pH Modification Using Acidifying/Alkalizing Agents

The most direct and effective way to change the pH_M_ is to add acids or bases into the formulations. The pH_M_ change might compromise the drug release from the formulations and the drug permeation, hence improving the drug absorption at the oral mucosa. Suitable pH shifts might increase the drug solubilities, despite them being poorly soluble in human saliva at the physiological pH. Previous studies have shown that the addition of organic acids leads to a significant increase in the dissolution of dapoxetine hydrochloride (DPX) particles in phosphate buffer at pH 6.8 (37 ± 0.5 °C) due to the pH-dependent solubility of DPX and the low pH_M_ around the drug particles [62,63]. In addition, the enhanced pharmacokinetic performance of DPX via the buccal films with organic acids in male Wistar rats was observed compared to that of the marketed DPX oral tablet (Priligy^®^) [63]. Another study showed that the addition of citric acid led to a faster nicotine release from buccal matrix tablets, whereas it inhibited nicotine permeation across the esophageal mucosa. Conversely, the permeation of nicotine was enhanced in the oral cavity due to the incorporation of magnesium hydroxide, despite the nicotine release being retarded [64]. Nicotine is a weak base (p*K*_a_ 3.04 and 7.84), the faster release of nicotine is due to the acidic pH_M_ created by the addition of citric acid. However, the released nicotine did not readily permeate across the mucosa. The ionized nicotine at an acidic pH has a low lipophilicity, which could lead to a limited permeation for the lipid-rich mucosal membrane [65,66]. Therefore, both the solubility and lipophilicity should be taken into consideration for a drug with pH-dependent solubility and lipophilicity.

### 6.2. Microenvironmental pH Modification Using Buffering Agents

The pH_M_ in the vicinity of formulations at the oral mucosa is generally affected by the release of the ingredients (particularly the acidic and basic ingredients) from the formulations. The pH_M_ changes over time, along with the ingredients released upon dissolution. To maintain the suitable pH_M_ and achieve optimal drug absorption at the oral mucosa, buffer agents are incorporated in formulations. The addition of buffering agents can form a buffer system in and around the matrix of the formulations and prevent the pH_M_ from changing. This method was demonstrated to be effective in some cases. Phosphate buffer and borate buffer were used in methylcellulose-based gels to create pH_M_ 7.4, 8.5, 9.0 and 9.5 for the buccal delivery of metoprolol in Göttingen minipigs in a previous study. A higher buccal absorption of metoprolol from the gels was observed at higher pH values, and the absolute bioavailability of metoprolol via buccal dosing was significantly higher compared to that via oral administration [29]. In this study, the metoprolol release from the gels might be similar, and the pH has little effect on metoprolol release, because the concentration of methylcellulose used in the gels was the same (1%, *w*/*v*) and metoprolol had already been dissolved in the gels. Metoprolol permeability across the buccal mucosa is the rate-limit step for the buccal absorption of metoprolol. Furthermore, metoprolol with p*K*_a_ 9.56 [85] has a pH-dependent lipophilicity and permeability in vitro and ex vivo [29,86]. Thus, the pH has a crucial influence on the buccal absorption of metoprolol incorporated in gels. In another work [27], a concept of pH_max_ was introduced to improve the sublingual delivery of weak base compounds. To verify the applicability of this concept, propranolol with a pH-dependent solubility and a pH-permeability across porcine sublingual mucosa was chosen as the model drug, and disodium hydrogen phosphate (buffering agent) was added into sublingual propranolol tablets to make the saliva pH (pH_M_) close to pH_max_ when the tablets dissolved in saliva under the tongue. The buffered sublingual propranolol tablet and the marketed tablet (Inderal^®^ which cannot achieve pH_max_) were administered sublingually, using the same procedure in eight healthy male volunteers. The buffered tablet led to a significantly higher plasma propranolol concentration than the marketed tablet at 10–30 min, at which point the drug release profiles were similar between the two formulations [27]. Therefore, pH_M_ modification using buffering agent is also a potential strategy to improve the buccal/sublingual delivery of drugs with pH-dependent solubility and/or permeability.

### 6.3. Microenvironmental pH Modification Using Effervescence

Formulations with effervescence generally contain an alkaline agent (e.g., sodium carbonate and sodium bicarbonate) and an acid that is capable of inducing the effervescence reaction during the dissolution [87]. The carbonic acid produced from the chemical reaction could decrease the pH_M_ and rapidly convert to water and carbon dioxide. The tablet using an effervescence reaction (containing citric acid and bicarbonate) was employed to enhance the absorption of fentanyl at the buccal mucosa [54,56,88]. A dynamic shift in the pH_M_ (pH was decreased and subsequently be increased) occurred in the microenvironment between the tablet and the buccal mucosa, and the pH_M_ shift might be the main factor for the enhanced buccal absorption of fentanyl. The initial decrease in the pH, caused by the carbonic acid and release of citric acid from the tablet, facilitated the release of fentanyl from the tablet. The pH subsequently increased due to the dissociation of carbonic acid (into CO_2_ and water) and the dissipation of the CO_2_, which favored the formation of unionized fentanyl. The unionized fentanyl can move across the lipid-rich oral mucosal membrane with greater ease than the ionized fentanyl [54,56,88]. In addition, the thinning or tripping of the mucus layer, the disruption of the epithelial barrier, and/or the increased membrane hydrophobicity caused by the CO_2_ might be key factors leading to a high buccal/sublingual absorption of drugs [89,90]. In addition, effervescence led to a faster release of buspirone hydrochloride in vitro from buccal discs and a significantly higher bioavailability of buspirone hydrochloride when compared to a conventional buccal formulation without effervescence [91].

## 7. Challenges

Despite pH_M_ modification being a potential strategy to enhance drug absorption via buccal/sublingual formulations, the main challenge is the possible local irritation to the oral mucosa and tooth erosion due to the pH shift in the oral cavity. Studies have shown that a low pH (<5.5) in the mouth can cause the erosion of dentine and enamel [92,93]. Damage was not observed in the buccal mucosa, tongue mucosa and salivary glands of rats when drinking water with an acidic pH (pH 5.0) [94]. However, acute effects (e.g., irritation and toxicity) on the oral mucosa may be expected at a pH lower than 2.5 [95]. In fact, the damage caused by the addition of pH modifiers in formulations on the oral mucosa and teeth relates to a combination of factors, i.e., the pH, total acid content, method of delivery and duration of contact [95]. These factors should be taken into consideration, and the irritation of pH_M_ modifying buccal/sublingual formulations to the oral mucosa should be investigated before their clinical use. 

For drug delivery purposes, the mucoadhesion implies the attachment of a drug delivery system to the mucous coat on the surface of the target tissues [96,97,98]. Several simultaneous interfacial interactions are involved in the complex mucoadhesion process, and the interactions can occur through covalent bonds or supramolecular interactions (including hydrogen bond and hydrophobic/electrostatic interactions) [96,99,100,101]. In the pH_M_ modifying buccal/sublingual formulations, polymers with a fairly good mucoadhesion are applied to extend the retention time of the formulations on the oral mucosa. Despite the potential of pH_M_ modification to enhance drug absorption at the oral mucosa, the shift in the pH_M_ might change the mucoadhesion and, hence, reduce the retention time of the formulations. The reason for this adverse effect may be that the dissociation of the functional groups on the polypeptide backbone in the mucin is pH-dependent. The pH_M_ shift might influence the charge on the surface of the mucus and, hence, reduce the interactions between the mucus and the formulations [96,97]. Additionally, CO_2_ produced by the effervescence could hinder the process of mucoadhesion. It might also create a porous contact surface in mucoadhesive formulations, leading to a poor interaction between the formulations and the mucin [91].

pH modification by adding pH modifiers might accelerate drug degradation in mucoadhesive formulations during the application and storage. The reason is that the pH is one of the most important factors affecting drug hydrolysis [102]. In addition, there are many reports in the literature referring to the effect of pH modifiers on the chemical stability of drugs in pharmaceutical solids [8,103,104,105,106,107]. Inappropriate pH modification (unsuitable pH modifier and the use of excessive pH modifiers) may lead to poor manufacturability. A previous study has reported that the type of acids affected the manufacturability of dipyridamole granules. During the process of wet granulation, the acids with different solubilities might partially dissolve in the binder solution, affecting the formation of the granules. The granules with toluenesulfonic acid monohydrate were formed as fine granules. However, the granules with maleic acid were problematic [108].

Compendial (USP and Ph. Eur.) methods and apparatuses for conventional tablets and gels have been widely employed for drug dissolution and the release testing of buccal/sublingual dosage forms in vitro [109,110,111,112]. However, most of these methods do not sufficiently mimic the physiological conditions present in the oral cavity. Relatively large volumes of dissolution medium with stirring have been used in these methods to create sink conditions, particularly in the cases of poorly water soluble drugs. Rapidly disintegrating formulations (e.g., mucoadhesive buccal/sublingual creams and gels) would disintegrate and/or dissolve within a few minutes upon contact with large volumes of dissolution medium during stirring. Our previous study shows that the combination of the Franz diffusion cell method and the UV/Vis imaging-method provided beneficial information to the drug release study of pH_M_ modifying mucoadhesive buccal films [59]. However, the physical light blocking caused by the polymeric matrix of the film led to a short measuring time (10 min). In the future, the UV/Vis setup should be further developed to solve the problems caused by the soluble matrix-forming polymers with low viscosity grades in buccal/sublingual formulations.

## 8. Conclusions

Microenvironmental pH (pH_M_) modification has attracted considerable interest as an effective strategy to promote systemic drug delivery across the oral mucosa, particularly for drugs with pH-dependent solubility and/or pH-dependent lipophilicity. In general, optimizing drug absorption at the oral mucosa using pH_M_ modification is considered to be a compromise between drug solubility and drug lipophilicity. A successful pH_M_ modification should ensure that the drugs are sufficiently soluble and lipophilic. This modification can facilitate drug releasing from the formulations and, subsequently, facilitate absorption across the oral mucosal membrane. The application of suitable pH modifiers and the monitoring of the pH_M_ during the dissolution of formulations are crucial for the design and development of pH_M_ modifying buccal/sublingual formulations. Among the pH_M_ measurement methods, the computer-enhanced color images of pH paper method and the UV/Vis imaging method provided beneficial information on the pH_M_ during the dissolution process. Buccal/Sublingual dosage forms using different pH_M_ modification methods (including using acidifying/alkalizing agents, buffering agents and effervescence, respectively) have been found to enhance drug bioavailability in animal models and human subjects. However, inappropriate pH_M_ modification might cause some problems, such as local irritation to the oral mucosa, tooth erosion, poor mucoadhesion and poor drug stability. Generally, pH_M_ modification in buccal/sublingual dosage forms has great potentials to enhance drug absorption. A better understanding of the theories and challenges of pH_M_ modification will be helpful for the future design of pH_M_ modifying buccal/sublingual formulations for systemic drug delivery. 

## Figures and Tables

**Figure 1 pharmaceutics-15-00637-f001:**
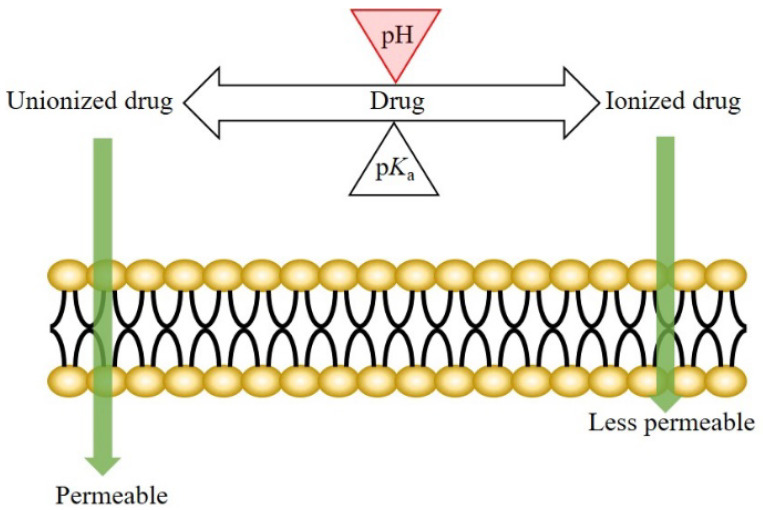
Illustration of pH-partition theory.

**Figure 2 pharmaceutics-15-00637-f002:**
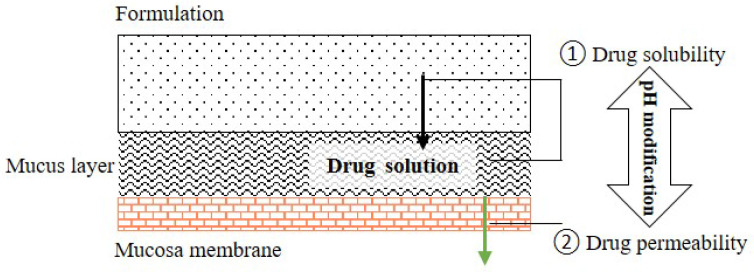
Schematic drawing of drug permeation across oral mucosa from the buccal/sublingual formulation.

**Figure 3 pharmaceutics-15-00637-f003:**
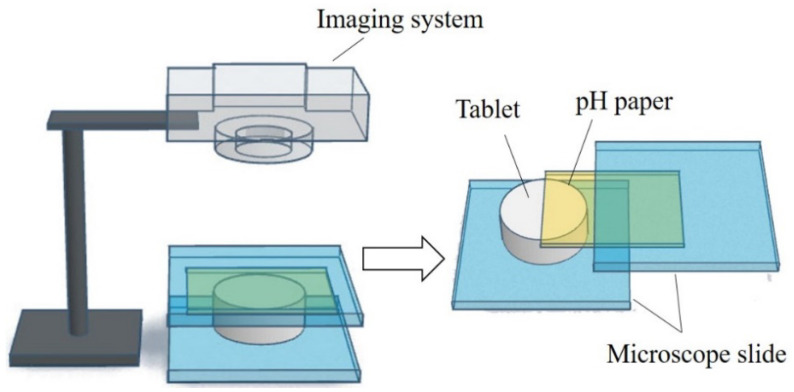
Schematic view of the setup of the computer-enhanced color images method.

**Figure 4 pharmaceutics-15-00637-f004:**
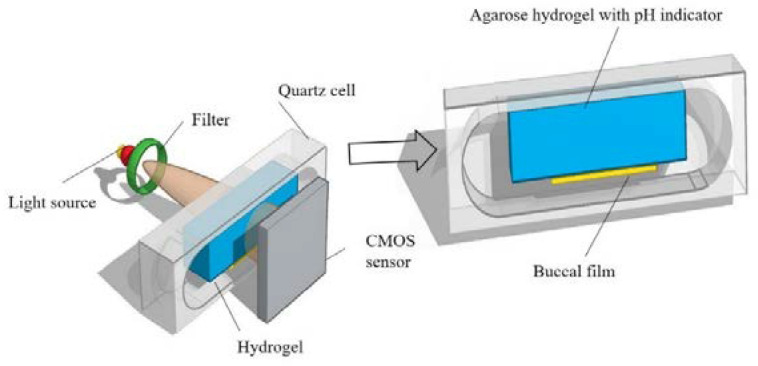
Schematic view of the UV/V is imaging setup, reprinted with permission from [58], copyright of ©2020 Elsevier.

**Figure 5 pharmaceutics-15-00637-f005:**
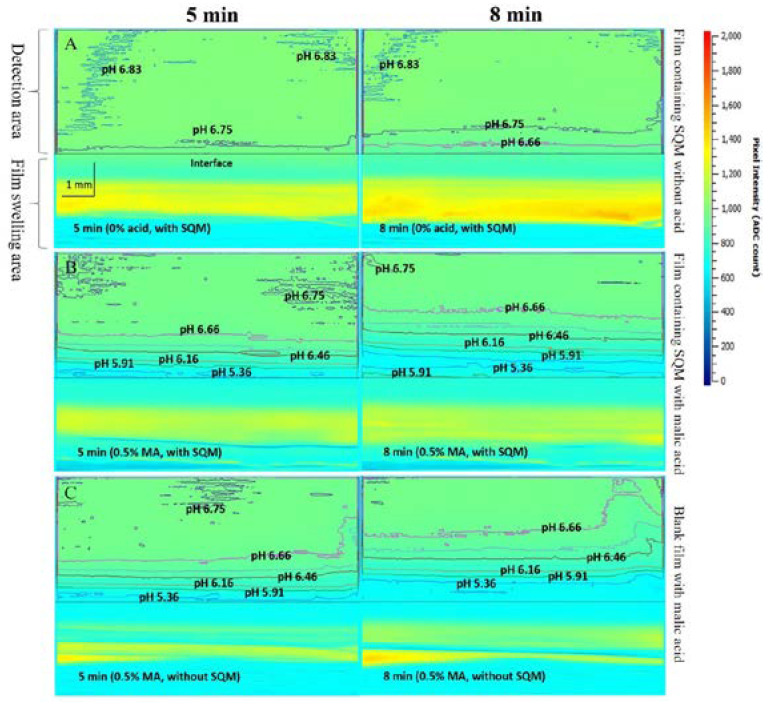
Selected absorbance images for the buccal films with agarose hydrogel (buffer solution) containing bromothymol blue at 610 nm. The difference of color indicates different absorbance, and the contours represent the iso-absorbance/iso-pH line. A. Film containing saquinavir mesylate (SQM) without acids. B. Film containing SQM with malic acid (MA). C. Film without SQM containing MA; 0.5% is the percentage of MA (*w*/*v*) dissolved in water during the film preparation, reprinted with permission from [58], copyright of ©2020 Elsevier.

**Table 1 pharmaceutics-15-00637-t001:** Selected buccal/sublingual drug products approved by FDA for systemic drug delivery.

Drug Name	Active Ingredients	Molecular Weight (g/mol)	Dosage Form	Administration Route	Company	Approval Date
Metandren^®^	Methyltestosterone	302.5	Tablet	Buccal, Sublingual	Novartis, Basel, Switzerland	-
Sorbitrate^®^	Isosorbide dinitrate	236.1	Tablet	Sublingual	Astrazeneca, Cambridge, UK	-
Isuprel^®^	Isoproterenol hydrochloride	247.7	Tablet	Sublingual, Rectal	Sanofi-Aventis US, Bridgewater (NJ), U.S.	1948
Hydergine^®^	Ergoloid mesylates	-	Tablet	Sublingual	Novartis, Basel, Switzerland	1953
Isordil^®^	Isosorbide dinitrate	236.1	Tablet	Sublingual	Biovail, Ontario, Canada	1961
Dentipatch^®^	Lidocaine	234.3	Film	Buccal	Noven, Miami (FL), U.S.	1996
Nitrostat^®^	Nitroglycerin	227.1	Tablet	Sublingual	Upjohn, Hastings (MI), U.S.	2000
Suboxone^®^	Buprenorphine hydrochloride; Naloxone hydrochloride	504.1, 363.8	Tablet	Sublingual	Indivior, Richmond (VA), U.S.	2002
Subutex^®^	Buprenorphine hydrochloride	504.1	Tablet	Sublingual	Indivior, Richmond (VA), U.S.	2002
Striant^®^	Testosterone	288.4	Tablet	Buccal	Auxilium Pharms, Centerbrook (CT), U.S.	2003
Fentora^®^	Fentanyl citrate	528.6	Tablet	Buccal, Sublingual	Cephalon, Frazer (PA), U.S.	2006
Edluar^®^	Zolpidem tartrate	764.9	Tablet	Sublingual	Mylan Speciality Lp, Basking Ridge (NJ), U.S.	2009
Saphris^®^	Asenapine maleate	401.8	Tablet	Sublingual	Allergan, Dublin, Ireland	2009
Onsolis^®^	Fentanyl citrate	528.6	Film	Buccal	BioDelivery Sciences International, Raleigh (NC), U.S.	2009
Suboxone^®^	Buprenorphine hydrochloride; Naloxone hydrochloride	504.1, 363.8	Film	Buccal, Sublingual	Indivior, Richmond (VA), U.S.	2010
Intermezzo^®^	Zolpidem tartrate	764.9	Tablet	Sublingual	Purdue Pharma, Stamford (CT), U.S.	2011
Abstral^®^	Fentanyl citrate	528.6	Tablet	Sublingual	Sentynl Theraps, Solana Beach (CA), U.S.	2011
Zubsolv^®^	Buprenorphine hydrochloride; Naloxone hydrochloride	504.1, 363.8	Tablet	Sublingual	Orexo US Inc., Morristown (NJ), U.S.	2013
Bunavail^®^	Buprenorphine hydrochloride; Naloxone hydrochloride	504.1, 363.8	Film	Buccal	BioDelivery Sciences International, Raleigh (NC), U.S.	2014
Belbuca^®^	Buprenorphine hydrochloride	504.1	Film	Buccal	BioDelivery Sciences International, Raleigh (NC), U.S.	2015
Dsuvia^®^	Sufentanil citrate	578.7	Tablet	Sublingual	Acelrx Pharmaceuticals, Hayward (CA), U.S.	2018
Nocdurna^®^	Desmopressin acetate	1129.3	Tablet	Sublingual	Ferring Pharmaceuticals, Saint-Prex, Switzerland	2018
Cassipa^®^	Buprenorphine hydrochloride; Naloxone hydrochloride	504.1, 363.8	Film	Sublingual	Teva Pharmaceuticals, Tel Aviv, Isreal	2018
Kynmobi^®^	Apomorphine hydrochloride	303.8	Film	Sublingual	Sunovion Pharmaceuticals, Marlborough (MA), U.S.	2020
Igalmi^®^	Dexmedetomidine	200.28	Film	Sublingual	BioXcel Therapeutics, New Haven (CT), U.S.	2022

**Table 2 pharmaceutics-15-00637-t002:** Physicochemical properties of transmucosal delivered drugs.

Drug	Molecular Weight (g/mol)	p*K*a *	Log P	Water Solubility (mg/mL)	Relevant Literature
Fentanyl	336.47	8.99	4.05	0.74	[54,55,56]
Naftopidil	392.50	7.35	3.65	0.07	[57]
Saquinavir	670.84	8.47	3.80	0.00247	[58,59,60,61]
Dapoxetine hydrochloride	305.41	8.96	4.75	0.00084	[62,63]
Nicotine	162.23	8.58	1.17	93.3	[64,65,66]
Metoprolol	267.36	9.67	2.15	0.402	[29]
Propranolol	259.34	9.42	3.48	61.7	[27]

* p*K*_a_ values recorded in DrugBank database might be different from those reported in the literatures referenced in this review.

**Table 3 pharmaceutics-15-00637-t003:** Selected pH modifiers and the maximum potency used in the drug products approved by FDA.

pH Modifiers	Maximum Potency Per Unit Dose (mg/Unit)	
	Oral Tablet/Capsule/Podwer	Buccal or Sublingual Tablet/Film
Acetic Acid	0.36	-
Adipic Acid	-	-
Ammonia	6.03	-
Ammonium Carbonate	-	-
Ammonium Chloride	10.67	-
Diammonium Phosphate	0.4	0.2
Boric Acid	-	-
Calcium Carbonate	550	145.7
Calcium Hydroxide	35	-
Calcium Lactate	-	-
Calcium Phosphate, Tribasic	333.3	99.2
Citric Acid Monohydrate	914	30
Citric Acid, Anhydrous	839	30
Diethanolamine	-	-
Fumaric Acid	150	-
Glycine	200	-
Hydrochloric Acid	1.72	-
Alpha-Lactalbumin	-	-
Lactic Acid	44	-
Lysine Hydrochloride	-	-
Maleic Acid	4	-
Malic Acid	315	-
Methionine	NA	-
Monoethanolamine	1	-
Monosodium Glutamate	-	-
Nitric Acid	-	-
Phosphoric Acid	1	-
Potassium Bicarbonate	500	8
Potassium Citrate	NA	-
Potassium Hydroxide	25.6	-
Potassium Metaphosphate	NA	-
Potassium Phosphate, Dibasic	30	-
Potassium Phosphate, Monobasic	25	-
Propionic Acid	-	-
Racemethionine	-	-
Sodium Acetate	-	-
Sodium Bicarbonate	1600	42
Sodium Borate	-	-
Sodium Carbonate	430	30
Sodium Citrate	1900	19.5
Sodium Hydroxide	60	1.18
Sodium Lactate Solution	-	-
Sodium Phosphate, Dibasic	600	4.07
Succinic Acid	125.3	-
Sulfuric Acid	-	-
Tartaric Acid	96	1.5
Trolamine	-	-

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
