# Peer review of "Microenvironmental pH Modification in Buccal/Sublingual Dosage Forms for Systemic Drug Delivery"

_pharmaceutics, 2023, doi:10.3390/pharmaceutics15020637_

Round 1
Reviewer 1 Report
In this manuscript, the use of pH modifiers in buccal dosage forms is reviewed.
General and specific points for revision are reported below.
Title and throughout the manuscript: the Authors refer to oral transmucosal dosage forms while buccal administration seems to be concerned.
Line 36: “mucus layer” should be omitted as permeation through the aqueous-rich mucus layer would require hydrophilicity rather than lipophilicity characteristics.
Line 38 and throughout the manuscript: it would be “dissolution” rather than “release”.
Line 201: “The selected drugs” should be changed to “The selected drug products”.
Table 1, caption: “..formulations approved by FDA” should be changed to “..drug products approved by FDA”.
Table 1, footnote: asterisks are misleading as they do not refer to any specific content in the table. Perhaps this information should be reported in the text.
Line 2015: “poorly soluble and/or permeable”?
Figure 3 is not strictly needed. The physico-chemical characteristics of drug candidates for buccal administration are listed in the text, as well as the properties that make drugs eligible for pH modification. The mechanism of pH modification has already been explained. The reprinted image is small in size and of low quality, and “optimal” should not be used in the lack of statistical optimization.
Table 2, caption: “drugs” should be changed to “drug products”.
Table 2, upper label: Would “power” read “powder”?
Table 2, upper label: “per unit” should be changed to “per unit dose”.
Table 2, footnote: asterisks are not necessary.
Line 243 and throughout the manuscript: the use of “matrix” should be avoided as it may suggest that prolonged-release formulations were dealt with
Figure 4: was this figure reprinted or adapted from a previous paper? Please mention the source and permission granted, if any.
Lines 310-311: it has been stated above (page 7 of 20) that pH modifiers can be divided into three categories.
Line 320: “consequently enhance drug permeation” should be left out: as discussed below (and also previously), the effect of pH modification on dissolution and mucosal permeation of drugs is often opposite.
Lines 362-65: what route of administration was the marketed product intended for? A higher bioavailability from sublingual versus oral tablets would not be surprising regardless of the presence or absence of pH modifiers.
Lines 371-72: in the effervescence reaction, carbon dioxide is produced from unstable carbonic acid.
Lines 383-84: opening of tight junctions would hardly result in increased hydrophobicity of the mucosa.
Lines 385-88. Did the Authors of the reviewed paper clarify to what extent the increased bioavailability of effervescent discs was due to faster disintegration and/or to pH modification?
Line 424: the concept of “overdose” would not apply to pharmaceutical excipients.
Lines 424-25: how would “inappropriate pH modification” relate to “poor manufacturability”? What do the Authors specifically mean by “poor manufacturability”?
The English language would benefit from a thorough revision. By way of example, please see:
-line 54, “desire”
-lines 59-61, passage to be checked and rephrased
-line 74, “an”
-lines 119-22, passage to be checked and rephrased
-lines 148-50, passage to be checked and rephrased
-line 195, “allow”
-lines 228-230 “tranmucosal”
-lines 245-48, passage to be checked and rephrased
- line 252, “has been”
- line 320, “shits”
-lines 406-08, unclear passage
- line 418, redundant wording
-lines 420-21, passage to be checked and rephrased
-lines 148-50, passage to be checked and rephrased
Author Response
Reviewers' comments:
Reviewer #1: In this manuscript, the use of pH modifiers in buccal dosage forms is reviewed.
General and specific points for revision are reported below.
- Title and throughout the manuscript: the authors refer to oral transmucosal dosage forms while buccal administration seems to be concerned.
RESPONSE: Thank you so much for the good comment. Oral transmucosal dosage forms mainly include oral mucoadhesive dosage forms (e.g. buccal/sublingual tablet, film, patch and gel etc.) and oral non-mucoadhesive dosage forms (e.g. gum, spray, powder etc.). Microenvironmental pH modification is generally applied in oral mucoadhesive dosage forms. To make it more specific, we have replaced the term, "Oral transmucosal dosage forms/formulations" with "oral mucoadhesive dosage forms/formulations" in the tittle and thorough the manuscript. Additionally, the non-mucoadhesive dosage forms have been deleted in Table 1. Actually, the sublingual administration has also be introduced throughout the manuscript.
- Line 36: “mucus layer” should be omitted as permeation through the aqueous-rich mucus layer would require hydrophilicity rather than lipophilicity characteristics.
RESPONSE: Thank you for your comment. "mucus layer" has be omitted accordingly. (see line 36 in the revised version)
- Line 38 and throughout the manuscript: it would be “dissolution” rather than “release”.
RESPONSE: Thank you for the comment. "Drug release" refers to the processes by which drug molecules are transferred from their initial position in a drug delivery system to the outer surface and, in turn, as solutes, into the release medium (https://link.springer.com/referenceworkentry/10.1007/978-3-030-84860-6_32, https://www.tandfonline.com/doi/full/10.1517/17425241003602259). "Drug dissolution" is the process in which a substance forms a solution. According to the USP and opinion of some experts working in the area of drug dissolution/release characteristics of pharmaceutical dosage forms, the two terms can be used interchangeably (https://www.usp.org/small-molecules/dissolution, https://drug-dissolution-testing.com/blog/files/dissomedia.pdf). FIP and AAPS Dissolution working group published two position papers on dissolution/drug release testing for novel/special dosage forms (https://link.springer.com/article/10.1208/pt040107, https://link.springer.com/article/10.1208/s12249-011-9634-x). The experts from FIP and AAPS considered that it is customary to refer to the test as a "dissolution" test for orally administered immediate release (IR) solid drug products, since the drug dissolves rapidly in the test medium. For non-oral dosage forms, the test is referred to preferably as a "drug release" or "in vitro release" test procedure. As novel/special dosage forms exhibit significant differences in formulation design, which in turn leads to very different physicochemical and release characteristics. Generally, the oral mucoadhesive dosage forms contain mucoadhesive polymers, which will not disintegrate immediately when wetted by the human saliva/the dissolution medium. The dosage forms will stay in the saliva/dissolution medium for a certain time period (the time period is defined as retention time). During the dissolution/release process, the dissolved drugs diffuse from their initial position in the oral mucosal dosage forms to the outer surface, and then into the dissolution medium. The drug dissolution/release process might be different form the conventional IR solid dosage forms. Thus, we think that the term "release" might be preferable in this manuscript.
- Line 201 (line 197-199 in the revised version): “The selected drugs” should be changed to “The selected drug products”.
RESPONSE: "transmucosal delivered drugs" has been changed to "mucoadhesive drug products", "The selected drugs" has been changed to "The selected drug products". (see line 197-199 in section 4.1).
- 5. Table 1, caption: “..formulations approved by FDA” should be changed to “..drug products approved by FDA”.
RESPONSE: The caption of Table 1 has been revised accordingly (line 201).
- Table 1, footnote: asterisks are misleading as they do not refer to any specific content in the table. Perhaps this information should be reported in the text.
RESPONSE: Thank you for the comment. The footnote of Table 1 has been deleted and the related information has been added in the text accordingly.
- Line 2015 (line 204-205 in the revised version): “poorly soluble and/or permeable”?
RESPONSE: Thank you for the question. The line 215, "which might have pH-dependent solubility and/or Ph-dependent lipophilicity." has been changed to "indicating that their solubility and/or permeability across the lipid-rich epithelium are pH-dependent." (line 204-205)
- 8. Figure 3 is not strictly needed. The physico-chemical characteristics of drug candidates for buccal administration are listed in the text, as well as the properties that make drugs eligible for pH modification. The mechanism of pH modification has already been explained. The reprinted image is small in size and of low quality, and “optimal” should not be used in the lack of statistical optimization.
RESPONSE: Figure 3 has been deleted accordingly.
- Table 2, caption: “drugs” should be changed to “drug products”. Table 2, upper label: Would “power” read “powder”? Table 2, upper label: “per unit” should be changed to “per unit dose”.
Table 2, footnote: asterisks are not necessary.
RESPONSE: The caption of Table 2 has been revised accordingly. Upper labels in Table 2, "power" and "per unit" have been changed to "powder" and "per unit dose". The footnote of Table 2 has been deleted.
- Line 243 and throughout the manuscript: the use of “matrix” should be avoided as it may suggest that prolonged-release formulations were dealt with
RESPONSE: Thank you for the comment. I agree with your point that the word "matrix" may suggest prolonged-release formulations in the manuscript. Matrix-based formulations are one of the popular methods to modify drug release behaviour for the oral drug administration. Generally, polymeric excipients like cellulose derivatives (e.g. HPMC), Kollidon SR, acrylic acid polymers (e.g. Eudragits and Carbopols) and polymers of natural origin (e.g. carrogens, chitosan and alginates) are matrix formers for the oral controlled drug delivery systems. Some of the polymers are also used to form matrix in oral transmucosal formulations (https://www.eurekaselect.com/article/46191). The matrix-based oral transmucosal delivery systems have also been studied recent years (https://link.springer.com/article/10.2165/00137696-200402030-00003). Thus, I think it would be fine to use the word "matrix" in this manuscript.
- Figure 4: was this figure reprinted or adapted from a previous paper? Please mention the source and permission granted, if any.
RESPONSE: No, it was not. The previous paper only introduced the detailed information of the setup in the text. We drew this figure according to the information.
- Lines 310-311 (line 317-321 in the revised version): it has been stated above (page 7 of 20) that pH modifiers can be divided into three categories.
RESPONSE: Thank you for the comment. The lines has been rephrased. (See line 317-321)
- Line 320 (line 324-325 in the revised version): “consequently enhance drug permeation” should be left out: as discussed below (and also previously), the effect of pH modification on dissolution and mucosal permeation of drugs is often opposite.
RESPONSE: Thank you for the comment. The effect of pH modification can compromise drug release and permeation hence improve drug absorption at oral mucosa. The lines related in the manuscript have been rephrased. (line 324-325, section 6.1)
- Lines 362-65: what route of administration was the marketed product intended for? A higher bioavailability from sublingual versus oral tablets would not be surprising regardless of the presence or absence of pH modifiers.
RESPONSE: Thank you for the comment. The marketed product, Inderal® was intended for oral administration route. However, both of the buffered and Inderal® tablets are applied through sublingual administration route using the same procedure (placing the tablet under the tongue and not to swallow any tablet or saliva until 15 min, at which time the saliva containing the dissolved propranolol was swallowed). Besides, both buffered and Inderal tablets were completely dissolved sublingually at about 15 min. Furthermore, the propranolol plasma concentration following buffered tablets was 1–2 folds higher than Inderal® tablet at 10–30 min, when the drug release profiles were similar between the two formulations. Therefore, this study indicates pH modification is a major factor contributing to the significantly faster sublingual absorption and higher propranolol plasma concentration. The related lines has been rephrased and more detailed information on this study has been added (line 367-371 in the revised version).
- Lines 371-72: in the effervescence reaction, carbon dioxide is produced from unstable carbonic acid.
RESPONSE: Thank you for the comment. The related lines have been revised (line 375-379).
- Lines 383-84: opening of tight junctions would hardly result in increased hydrophobicity of the mucosa.
RESPONSE: The lines may lead to misunderstanding. The lines have been revised (line 388-390).
- Lines 385-88. Did the Authors of the reviewed paper clarify to what extent the increased bioavailability of effervescent discs was due to faster disintegration and/or to pH modification?
RESPONSE: No, the authors did not clarify that.
- Line 424: the concept of “overdose” would not apply to pharmaceutical excipients.
RESPONSE: Thank you for this comment. Theoretically, excipients are pharmacologically inactive materials and nontoxic. From this point, it is impossible to overdose on pharmaceutical excipients. But some pharmaceutical excipients show effects on drug transporters and/or metabolic enzymes, or cause side effects/adverse effects when inappropriate applied in formulations (such as overdose) (https://www.mdpi.com/1422-0067/21/21/8224, https://link.springer.com/article/10.1208/s12248-016-9928-8, https://pubmed.ncbi.nlm.nih.gov/3287089/). Zuccoti GV and Fabiano V introduced the safety issues caused by overdose of the excipient, ethanol to pediatric population (https://www.tandfonline.com/doi/full/10.1517/14740338.2011.565328). The pharmaceutical excipient, ascorbic acid is a typical pH modifier and/or antioxidant in formulations (https://www.tandfonline.com/doi/full/10.3109/10837450.2012.751408). Overdosing ascorbic acid is a risk factor for calcium oxalate nephrolithiasis (https://www.auajournals.org/doi/10.1016/S0022-5347%2817%2937521-3).
- Lines 424-25: how would “inappropriate pH modification” relate to “poor manufacturability”? What do the Authors specifically mean by “poor manufacturability”?
RESPONSE: Thank you for your questions. pH modifiers are usually added in formulations to achieve pH modification. The types and concentration of pH modifiers might affect the manufacturability of the formulations. Taniguchi et al. investigated the type of pH modifiers on the manufacturability of dipyridamole granule (DPG). The physicochemical properties (e.g. solubility) affected the manufacturability of the granule. During the wet granulation process, the tested pH modifiers might partly dissolve in the binder solution, affecting the formation of DPG. DPG with toluenesulfonic acid nonohydrate were formed as fine granules. Granules with maleic acid were formed as problematic granules (https://www.sciencedirect.com/science/article/pii/S0378517312005406?via%3Dihub). More detailed information has been added in the related lines to explain the relationship between pH modification and manufacturability. (line 428-434)
The English language would benefit from a thorough revision. By way of example, please see:
- -line 54, “desire”
RESPONSE: the word "desire" has been changed to "desired" (line 54).
- 21. -lines 59-61, passage to be checked and rephrased
RESPONSE: Lines 59-61 have been changed to "Generally, pH modifiers could modulate pHM in and in the vicinity of the mucoadhesive formulations and affect the drug ionization, hence influence the drug permeation across the oral mucosa." (Line 59-61 in the revised version)
- -line 74, “an”
RESPONSE: the word "an" in line 74 has been changed to "a". (line 77)
- -lines 119-22, passage to be checked and rephrased
RESPONSE: the lines 119-22 have been changed to "However, previous studies about the metroprolol buccal tablet and gel did not support the pHmax concept. A specific pH level (rather than pHmax) also led to the highest buccal absorption of metroprolol." (line 118-120)
- -lines 148-50, passage to be checked and rephrased
RESPONSE: The lines 148-50 "The human saliva has been reported to have a pH range of 6.2-7.6. The pH of saliva secreted from different regions differs, due to the different quantity and it’s the components." have been changed to "The human saliva has been reported to have a pH range of 6.2-7.6. The composition of saliva secreted from different regions differs, leading to different saliva pH." (line 146-148)
- -line 195, “allow”
RESPONSE: The word "allow" has been changed to "allows". (line 190)
- -lines 228-230 “tranmucosal”
RESPONSE: The word "tranmucosal" has been changed to "transmucosal" (line 229-231).
- -lines 245-48, passage to be checked and rephrased
RESPONSE: The lines "Several techniques have been applied to gain information on the pHM to investigate the relationship between pHM and drug dissolution behavior. pH indicating dyes have been used to determine the pHM within and around the tablets during hydration, this dye method only roughly estimates the pHM depending on the color change." has been changed to:
"Several techniques have been applied to gain information on the pHM, and to investigate the relationship between pHM and drug dissolution behavior. pH indicating dyes have been used to determine the pHM within and around the tablets during hydration. This method only roughly estimates the pHM according to the relationship between the pH level and the color of dyes." (line 245-250)
- - line 252, “has been”
RESPONSE: The term "has been" in line 252 has been changed to "have been"(line 254).
- - line 320, “shits”
RESPONSE: the word "shits" has been changed to "shifts" (line 325).
- -lines 406-08, unclear passage
RESPONSE: The lines has been rephrased (line 410-412).
- - line 418, redundant wording
RESPONSE: line 418, "It might also create a porous contact surface to mucoadhesive formulations, which may led to poor interaction between polymeric chains and glycoprotein chains of mucin" has been changed to " It might also create a porous contact surface to mucoadhesive formulations, leading to poor interaction between the formulations and the mucin" (line 421-423).
- -lines 420-21, passage to be checked and rephrased
RESPONSE: The lines "pH modification by adding pH modifiers in formulations might accelerate the rate of drug degradation in semisolid mucoadhesive formulations and solid formulations upon the dissolution. Because pH is one of the most important factors affect the stability of drugs degrading by hydrolysis in solution" has been changed to:
"pH modification by adding pH modifiers might accelerate drug degradation in mucoadhesive formulations during application or/and storage. Because pH is one of the most important factors affecting drug hydrolysis" (line 424-426)
- -lines 148-50, passage to be checked and rephrased
RESPONSE: See comment 24.
Reviewer 2 Report
very nice , well structured and complete review!
Previous work on combined dissolution-/permeation-testing is not sufficiently covered among the references.
Author Response
Reviewer #2:
very nice , well structured and complete review!
- Previous work on combined dissolution-/permeation-testing is not sufficiently covered among the references.
RESPONSE: Thank you for your good comment. Developing proper dissolution testing methods for oral mucoadhesive formulations with a short disintegrate time and retention time is still a challenge. The related references on dissolution testing methods have been added in the section 7 (line 435-449).
Reviewer 3 Report
The present review describes the challenges and possibilities with microenvironmental pH modification to improve systemic delivery from oral transmucosal formulations. Additionally, it provides a brief overview of the physiological environment in the oral cavity, the general concept of microenvironmental pH modification, as well as a discussion of the methods to measure and modify microenvironmental pH.
In general, the review clearly presents the topic and could be of relevance to other researchers working within the field of drug delivery to the oral cavity. The review provides appropriate figures supporting the text and making it easier to understand the topic. However, there is a few general and specific comments which should be addressed before the manuscript would be suitable for publication:
General concept comments:
1) In the introduction, some characteristics of the oral cavity and mucus is nicely summed up, providing a good overview of the physiological environment before proceeding into the discussion of possible formulation tools to be used to improve drug delivery. However, it is not clearly stated whether all of the numbers in the introduction refers to the physiological environment in humans? Or if some of them are reported from animals? Since most of the studies discussed in the other sections of the review are based on animal studies, it would be very relevant to address if the physiological parameters in the oral cavity is similar between humans and the laboratory animals used in the cited references. For example, it has been well-documented for the rest of the gastrointestinal tract, that pH, composition and buffer capacity vary significantly between species.
2) In general, the reference list contains a good selection of relevant references to cover the subject. However, there is a lack of recent literature (only 16/100 references are from within the last 5 years). Therefore, it is highly recommended that the others review the most recent literature in the field to make sure that no novel and relevant articles were omitted.
3) It is recommended to include the Henderson-Hasselbach equation in the review (section 4.2), as it has been done for the other relevant equations highlighted in the review (e.g. Section 2.1).
4) Since the * in the table footer do not refer to a specific cell in the two tables, it is recommended to move this information to the table caption.
5) Please elaborate on some of the studies cited in section 5.1, as it has been done in all other sections in the manuscript. It will make it easier for the reader to follow the argumentation if the cited work is briefly described in the review, and not only cited.
6) Please include relevant physical-chemical properties of all drugs discussed in the review, for example metropolol and propranolol.
Specific comments:
1) Line 191-193: This is a repetition of what has already been written in line 103-104.
2) Line 356-357: Please include the appropriate reference for the work discussed in the beginning of the sentence.
3) Several sentences are not completely clear, which makes it difficult to follow the argumentation. Therefore, it is recommended to revise/rephrase the following sentences:
a. Line 121-122: “…observed from the formulation created a pH led to the lowest solubility…”
b. Line 151-155 (from “despite” to “pH shifts”): Confusing sentence structure.
c. Line 196-201: Confusing sentence structure.
d. Line 233-235: Rephrase.
4) In general, there is a lot of places in the manuscript where it is difficult to understand the message due to mixed usage or lack of definite/indefinite articles (an, a, the) and/or plural (for example line 115, 117, 119, 182, 188, 248, 249, 310). Therefore, extensive editing of English language and style is recommended.
Author Response
Reviewer #3:
The present review describes the challenges and possibilities with microenvironmental pH modification to improve systemic delivery from oral transmucosal formulations. Additionally, it provides a brief overview of the physiological environment in the oral cavity, the general concept of microenvironmental pH modification, as well as a discussion of the methods to measure and modify microenvironmental pH.
In general, the review clearly presents the topic and could be of relevance to other researchers working within the field of drug delivery to the oral cavity. The review provides appropriate figures supporting the text and making it easier to understand the topic. However, there is a few general and specific comments which should be addressed before the manuscript would be suitable for publication:
General concept comments:
1) In the introduction, some characteristics of the oral cavity and mucus is nicely summed up, providing a good overview of the physiological environment before proceeding into the discussion of possible formulation tools to be used to improve drug delivery. However, it is not clearly stated whether all of the numbers in the introduction refers to the physiological environment in humans? Or if some of them are reported from animals? Since most of the studies discussed in the other sections of the review are based on animal studies, it would be very relevant to address if the physiological parameters in the oral cavity is similar between humans and the laboratory animals used in the cited references. For example, it has been well-documented for the rest of the gastrointestinal tract, that pH, composition and buffer capacity vary significantly between species.
RESPONSE: Thank you so much for the good comment. The species of the physiological environments described has been stated in the manuscript (line 35, 51, 55 in section 1 and line 156, 166, 171, 173, 176 in section 3).
2) In general, the reference list contains a good selection of relevant references to cover the subject. However, there is a lack of recent literature (only 16/100 references are from within the last 5 years). Therefore, it is highly recommended that the others review the most recent literature in the field to make sure that no novel and relevant articles were omitted.
RESPONSE: several recent literatures published recently related to mucoadhesive polymers and drug release testing methods for oral mucoadhesive formulations have been referred in the manuscript accordingly (reference 58, 59, 60, 61, 98, 101).
3) It is recommended to include the Henderson-Hasselbach equation in the review (section 4.2), as it has been done for the other relevant equations highlighted in the review (e.g. Section 2.1).
RESPONSE: Henderson-Hasselbach equation has been added in section 4.1 accordingly (line 215).
4) Since the * in the table footer do not refer to a specific cell in the two tables, it is recommended to move this information to the table caption.
RESPONSE: the footnote of the Table has been removed accordingly.
5) Please elaborate on some of the studies cited in section 5.1, as it has been done in all other sections in the manuscript. It will make it easier for the reader to follow the argumentation if the cited work is briefly described in the review, and not only cited.
RESPONSE: Some of the studies cited in section 5.1 have been introduced accordingly (line 265-270).
6) Please include relevant physical-chemical properties of all drugs discussed in the review, for example metropolol and propranolol.
RESPONSE: The relevant physical-chemical properties have been summarized in Table 2 accordingly.
Specific comments:
1) Line 191-193: This is a repetition of what has already been written in line 103-104.
RESPONSE: Lines 191-193 have been changed to "The extent to different pathways depends on the drug physicochemical properties." (line 188).
2) Line 356-357: Please include the appropriate reference for the work discussed in the beginning of the sentence.
RESPONSE: The reference 27 for the work has been added accordingly (line 362).
3) Several sentences are not completely clear, which makes it difficult to follow the argumentation. Therefore, it is recommended to revise/rephrase the following sentences:
- Line 121-122: “…observed from the formulation created a pH led to the lowest solubility…”
RESPONSE: The lines have been rephrased (line 118-120)
- Line 151-155 (from “despite” to “pH shifts”): Confusing sentence structure.
RESPONSE: The lines have been changed to "The flow of saliva with a buffer capacity resisting pH shifts could remove acidic and basic foods on the oral mucosa, maintaining the pH in the oral cavity near neutrality in a long term." (Line 149-151)
- Line 196-201: Confusing sentence structure.
RESPONSE: Line 196-201 has been changed to " Macromolecules are possible to be delivered via oral mucosa, e.g., buccal insulin spray (Generex Oral-lyn®) was approved by Food and Drug Administration (FDA) for treatment of patients under the Investigational New Drug (IND) program. However, the number of marketed oral transmucosal macromolecules is very small. Most of the marketed oral transmucosal delivered drugs are small molecules." (line 192-196).
- Line 233-235: Rephrase.
RESPONSE: Line 233-235: "Despite only a few pH modifiers in Table 2 were applied in the commercial oral transmucosal formulations approved by FDA, pH modifiers without local irritation and toxicity to oral mucosa could be potential choices." have been changed to:
"Currently, only a few pH modifiers in Table 2 were applied in the commercial oral transmucosal formulations approved by FDA. pH modifiers demonstrated without local irritation and toxicity to oral mucosa could also be potential choices for oral transmucosal formulations." (line 234-237).
4) In general, there is a lot of places in the manuscript where it is difficult to understand the message due to mixed usage or lack of definite/indefinite articles (an, a, the) and/or plural (for example line 115, 117, 119, 182, 188, 248, 249, 310). Therefore, extensive editing of English language and style is recommended.
RESPONSE: Thanks for your suggestion. Extensive editing has been done for this manuscript.
Round 2
Reviewer 1 Report
The line numbers indicating revisions do not correspond in the pdf file, making the referee’s task more difficult.
I do not agree with the use of “oral” in this context. Buccal administration is actually concerned, and changing “transmucosal” to “mucoadhesive” does not solve this issue.
I fully understand that “overdose” applies to ethanol used as an excipient, and may also apply to ascorbic acid. However, in this passage, “overdose” does not seem to refer to tolerability but rather to an excess amount of pH modifier that impacts manufacturability. Therefore, I would avoid the term.
The English wording should be double-checked. Please see for instance the change noted at lines 424-26, which is actually found at lines 466-70: “Because pH is one of the most important factors affecting drug hydrolysis.” cannot stand alone.
Author Response
Reviewer #1:
- The line numbers indicating revisions do not correspond in the pdf file, making the referee’s task more difficult.
RESPONSE: Thank you for your patience. We are sorry for the inconvenience. We submited a MS word file of this revised manuscript to the editor. Maybe, the MS word file was transformed into a pdf file automatically by the system or by the editor. There are some changes in the line numbers in the pdf file. I will contact the editor to solve this problem.
- I do not agree with the use of “oral” in this context. Buccal administration is actually concerned, and changing “transmucosal” to “mucoadhesive” does not solve this issue.
RESPONSE: Thank you for your meticulousness. I agree with your point. Actually buccal and sublingual dosage forms are introduced in this manuscript. To make it precise, the terms, "oral mucoadhesive" and "oral transmucosal" have been changed to "buccal/sublingual" throughout the manuscript.
- I fully understand that “overdose” applies to ethanol used as an excipient, and may also apply to ascorbic acid. However, in this passage, “overdose” does not seem to refer to tolerability but rather to an excess amount of pH modifier that impacts manufacturability. Therefore, I would avoid the term.
REPONSE: Thank you for your comment. The term, "overdose of pH modifiers" has been changed to "the use of excessive pH modifiers". (Sentence 4, Paragraph 3 in Section 7 - line 479-480)
- The English wording should be double-checked. Please see for instance the change noted at lines 424-26, which is actually found at lines 466-70: “Because pH is one of the most important factors affecting drug hydrolysis.” cannot stand alone.
RESPONSE: Thank you for this comment. The sentence, "Because pH is one of the most important factors affecting drug hydrolysis." has been changed to "The reason is that pH is one of the most important factors affecting drug hydrolysis." (Sentence 3, Paragraph 3 in Section 7 – line 475-477). The English wording has been checked again. Some other changes have been made based on the latest version are described as below:
1) The term, "at the surface of solid formulation" has been changed to "at the surface of a solid formulation". (Sentence 6, Paragraph 4 in Section 2.1 – line 102)
2) The term, "biological membrane" has been changed to "biological membranes". (Sentence 8, Paragraph 5 in Section 2.1 – line 112)
3) The term, "with pH paper" has been changed to "with the pH paper". (Sentence 4, Paragraph 1 in Section 5.2 – line 362)
4) The caption of Fig 3 has been changed. (line 325)
5) The caption of section 6.1, "Microenvironmental pH Modification Using Acidifying/alkalizing Agent" has been changed to "Microenvironmental pH Modification Using Acidifying/Alkalizing Agents".
6) The caption of section 6.2, "Microenvironmental pH Modification Using Buffering Agent" has been changed to "Microenvironmental pH Modification Using Buffering Agents".
7) The term, "pHM shift" has been changed to "The pHM shift". (Sentence 6, Paragraph 2 in Section 7 – line 491)
8) The term, "using acidifying/alkalizing agent, buffering agent" has been changed to "using acidifying/alkalizing agents, buffering agents". (Sentence 7, Paragraph 1 in Section 8 – line 517)
Round 3
Reviewer 1 Report
All points I raised have been addressed.